# The Societal Value of Vaccines: Expert-Based Conceptual Framework and Methods Using COVID-19 Vaccines as a Case Study

**DOI:** 10.3390/vaccines11020234

**Published:** 2023-01-20

**Authors:** Manuela Di Fusco, Diana Mendes, Lotte Steuten, David E Bloom, Michael Drummond, Katharina Hauck, Jonathan Pearson-Stuttard, Rachel Power, David Salisbury, Adrian Towse, Julie Roiz, Gabor Szabo, Jingyan Yang, Kinga Marczell

**Affiliations:** 1Health Economics and Outcomes Research, Pfizer Inc., New York, NY 10017, USA; 2Health & Value, Pfizer Co., Ltd., Tadworth KT20 7NS, UK; 3Office of Health Economics, London SW1E 6QT, UK; 4Department of Global Health and Population, Harvard T. H. Chan School of Public Health, Boston, MA 02115, USA; 5Centre for Health Economics, Alcuin A Block, University of York, Heslington, York YO10 5DD, UK; 6Department of Infectious Disease Epidemiology, Faculty of Medicine, School of Public Health, Imperial College London, London W2 1PG, UK; 7Department of Epidemiology and Biostatistics, School of Public Health, Imperial College London, London W2 1PG, UK; 8Health Analytics, Lane Clark & Peacock, London W1U 1DQ, UK; 9The Patients Association, PO Box 935, Harrow HA1 3YJ, UK; 10Programme for Global Health, Royal Institute of International Affairs, Chatham House, London SW1Y 4LE, UK; 11Evidence, Value and Access by PPD, Evidera, London W6 8BJ, UK; 12Evidence, Value and Access by PPD, Evidera, H-1113 Budapest, Hungary; 13Institute for Social and Economic Research and Policy, Graduate School of Arts and Science, Columbia University, New York, NY 10027, USA

**Keywords:** vaccines, COVID-19, health technology assessment, vaccine value, COVID-19 vaccination, Delphi, expert consensus, societal impact

## Abstract

Health technology assessments (HTAs) of vaccines typically focus on the direct health benefits to individuals and healthcare systems. COVID-19 highlighted the widespread societal impact of infectious diseases and the value of vaccines in averting adverse clinical consequences and in maintaining or resuming social and economic activities. Using COVID-19 as a case study, this research work aimed to set forth a conceptual framework capturing the broader value elements of vaccines and to identify appropriate methods to quantify value elements not routinely considered in HTAs. A two-step approach was adopted, combining a targeted literature review and three rounds of expert elicitation based on a modified Delphi method, leading to a conceptual framework of 30 value elements related to broader health effects, societal and economic impact, public finances, and uncertainty value. When applying the framework to COVID-19 vaccines in post-pandemic settings, 13 value elements were consensually rated highly important by the experts for consideration in HTAs. The experts reviewed over 10 methods that could be leveraged to quantify broader value elements and provided technical forward-looking recommendations. Limitations of the framework and the identified methods were discussed. This study supplements ongoing efforts aimed towards a broader recognition of the full societal value of vaccines.

## 1. Introduction

The dynamic changes in the social landscape brought by coronavirus disease 2019 (COVID-19) have shown the multiplicity of harms and broad impacts that infectious diseases may bring to society. The public health and societal impact of the COVID-19 pandemic and the associated economic and social fallout have been far-reaching [1,2,3,4,5]. The measures taken by governments to mitigate the pandemic, such as mass vaccination, social distancing, mask wearing, and lockdowns have also directly or indirectly affected individuals and their social interactions, health systems and the world economy [6,7,8,9].

Traditional health technology assessment (HTA) focuses on a narrow list of health-related effects, including reduction in mortality and morbidity, lowering healthcare costs and resource use, and, in certain HTA jurisdictions, increasing the productivity of patients and their caregivers [10,11,12].

A growing body of evidence advocates that vaccines have benefits and externalities that extend beyond these traditional categories; hence, they may be systematically undervalued in HTAs [13,14,15,16,17,18,19,20,21,22,23,24,25,26,27,28,29]. Over the past decade, several generic and vaccine-specific value frameworks have been developed using a broad societal perspective [13,14,15,16,17,18,19,20,21,22,23,24,25,26,27,28,29]. More recently, researchers concurred that the COVID-19 pandemic showed the limits of current HTAs and contributed to broadening the view of what constitutes vaccines’ value in HTA [13,17,22,24,28,30]. Bell et al. [13] identified gaps between the broader value elements included in early vaccination-specific value frameworks and the value elements generally considered in HTAs of vaccines in nine higher-income countries. The researchers found variations in the value elements considered, reflecting between-country differences in HTA methods and health policy focus. Some countries were found to have greater ability and willingness to close the gaps by incorporating broader value elements within their processes. The findings underpinned the development of a synthesised framework, called the ‘Broader Value of Vaccines’ (BRAVE) Framework, which represents a roadmap for advancing the consideration of the broader value of vaccines. [13].

While the broader value of vaccines is increasingly recognised and theoretically understood, there is limited research and consensus on the methods to use to measure broader value elements and integrate them meaningfully in economic evaluations and decision-making [26,29].

Building on the existing body of evidence, expert inputs, and the growing literature on the effects of COVID-19 vaccination, the aims of this study were threefold: (1) to create a consensus-based conceptual framework capturing the broader value of vaccination, (2) to gather expert recommendations and views on challenges related to analytical approaches measuring these benefits, and (3) to assess the applicability of the framework and the identified methods for evaluations of COVID-19 vaccines, with a focus on post-pandemic settings.

## 2. Materials and Methods

The study adopted a two-step approach combining a targeted literature review (TLR) and an expert elicitation exercise.

### 2.1. Targeted Literature Review (TLR)

The objectives of the TLR were: (1) to identify general and vaccine-specific value frameworks characterising vaccine value, (2) to describe the economic and societal outcomes impacted by COVID-19, and (3) to identify methods to include such impacts in HTA processes.

Considering the breadth of the research topics underpinning these objectives and the vastness of the COVID-19 literature, an all-encompassing systematic review would have covered almost the entirety of the vaccine literature. We therefore opted to leverage the latest research by Brassel et al. [17] and Bell et al. [13] to understand the current state of vaccine value assessments and to conduct targeted searches to identify relevant articles on the impact of COVID-19 and the effects of COVID-19 vaccination. The targeted searches were conducted between 14 December, 2021 and 7 April, 2022 using PubMed and Google Scholar. The TLR integrated two main types of evidence: observational studies and models, with the latter being especially helpful to identify relevant analytical approaches.

The latest vaccine frameworks and the evidence on economic and societal outcomes impacted by COVID-19 were analysed and synthesised in an ‘Impact Inventory’. A gap analysis was conducted to identify elements of the impact inventory that are not currently recognised in HTAs, with a focus on the UK and the US. Finally, available methodologies for their quantification were identified and summarised. An appraisal of the literature was conducted before the expert elicitation phase. The appraisal considered both the quality of the evidence linking the value elements to COVID-19 and/or vaccines, and the ability to incorporate the value elements into economic evaluations, expressed as the feasibility of monetising them (Appendix A).

### 2.2. Expert Elicitation

The purpose of the expert elicitation phase was fourfold: (1) to review and validate the TLR findings, (2) to leverage the ‘Impact Inventory’ to develop a consensus-based conceptual framework of broad vaccine value elements, (3) to seek consensus on the broader value elements of the framework to be prioritised in post-pandemic COVID-19 vaccine assessments, and (4) to gather recommendations and challenges regarding methods quantifying those prioritised value elements.

Similar to Bell et al., [13] the expert elicitation phase was based on a modified Delphi, a well-established technique combining existing evidence with multiple expert perspectives to identify gaps and priorities for research and decision making [31,32].

The expert elicitation was performed in three rounds (Figure 1) and leveraged a questionnaire developed based on the TLR findings (Appendix A). Round 1 was an individual expert elicitation. Round 2 and Round 3 expert elicitations were conducted during a virtual panel discussion (Panel 1). Panel 1 was followed by another virtual panel (Panel 2), which was a qualitative discussion to review the methods identified in the TLR. While the two panel discussions allowed group interaction, all elicitations were individual and anonymous.

Given the complexity and breadth of the research topics, the panel eligibility criteria included in-depth knowledge of HTA methods and decision making as well as expertise in applicable areas such as health economics, public health, immunisation, outcomes research, infectious disease epidemiology, HTA and health policy. A total of nine experts with comprehensive and authoritative knowledge engaged in the discussions and the individual expert elicitations. The panel included an expert in the direct elicitation of patients and provider benefit-risk trade-off preferences, and a patient advisory group representative. A distinguished expert in health economics acted as the moderator for both panels, and also contributed to the discussions while participating in the individual expert elicitation rounds as a facilitator.

The panel sample size was defined based on the BRAVE example [13], prior research and guidelines [31,32,33,34,35,36], and considerations of response rate, group dynamics, and stability of responses. A 70% level of agreement was considered appropriate according to the Delphi guidelines and prior Delphi studies with similar panel size [31,32,33,34,35,36,37,38,39]. Hence, our study aimed for this threshold with slight modifications in either direction as the number of responders changed slightly across rounds, as further detailed in the Results section.

#### 2.2.1. Round 1: Individual Expert Elicitation

In Round 1, each expert was provided with a summary of the TLR findings, including the ‘Impact Inventory’, and was invited to take part in a survey via email (Appendix A). The experts were asked to rate the value elements that, conceptually, were priorities for inclusion in the economic evaluations of COVID-19 vaccines, as well as the quality of evidence underpinning their inclusion. They also commented on the overall feasibility of their inclusion or the rationale for their exclusion in existing HTA frameworks. The questionnaire used a 5-point Likert scale, along with a few open-ended questions that asked the experts to state their view independently of the perspective recommended by the HTA bodies (Appendix A).

The experts then were asked if the methods identified in the TLR were suitable for the inclusion of broader value elements within economic evaluations of COVID-19 vaccines. Finally, they were asked about other potential methods and relevant examples and best practices.

Round 1 was followed by two half-day virtual panel discussions: Panel 1 in May 2022 and Panel 2 in June 2022.

#### 2.2.2. Panel 1 (Rounds 2 and 3)

Panel 1 focused on building an expert-informed conceptual framework encompassing broader vaccine value elements, and on gathering consensus on the value elements to be prioritised for inclusion in economic evaluations of COVID-19 vaccines, with a focus on post-pandemic settings.

During Panel 1, two main sets of results were presented to the experts: (1) the TLR findings, including the ‘Impact Inventory’ and a gap analysis versus existing HTA frameworks, and (2) the Round 1 survey results, analysed prior to the virtual discussion. The presentation of these results was followed by a moderated group discussion to reflect on Round 1 responses, debate interpretations and uncover differences in opinion. Following clarifications and discussion, the experts were invited to take part in Round 2. Via live online polls, Round 2 repeated the prioritisation process with a focus on the value elements for which consensus was not achieved in Round 1. A subsequent group discussion was conducted to discuss the new value elements that were prioritised. In Round 3, the experts were asked to select the top five value elements for which consensus was not reached during the prior rounds based on their relevance or priority for inclusion in economic evaluations. The experts collectively reviewed the resulting consensus, as well as a summary of the main topics and expert considerations that emerged from Panel 1.

#### 2.2.3. Panel 2

Panel 2 was a focused semi-structured technical discussion on the methods and challenges involved in capturing and quantifying the broader value elements prioritised during Panel 1. Panel 2 aimed to discuss challenges and recommendations for methodological approaches enabling the integration of the broader effects of vaccination in economic evaluations, with a focus on COVID-19 vaccines. There was no formal voting or prioritisation. The experts were asked to review existing analytical approaches, and to share examples and solutions to overcome some of the challenges associated with these approaches. The results of the discussion were qualitatively analysed based upon a content analysis of notes and transcripts. A descriptive summary was organised around each value element.

## 3. Results

### 3.1. Findings of the Targeted Literature Review

The TLR resulted in the identification of more than 15 value frameworks, with eight of them specific to vaccines, indicating significant progress and strong advocacy to advance vaccine value assessment using a broad societal perspective [13,15,16,17,18,19,20,21,22,23,24,25,27,28,29].

The TLR identified a relatively long list of value elements that are not captured in traditional vaccine value assessments [10,11,12]. A total of 30 identified value elements were described in an ‘Impact Inventory’ (Appendix A), further detailed in the next section.

The results of a gap analysis versus existing frameworks showed that several of the value elements are presented in different levels of detail and classification in various existing frameworks, as shown in Figure 2 and Appendix A. One of the latest vaccine-specific value frameworks, BRAVE [13], captured all the key categories of value attributed to vaccines in prior frameworks. As such, recognising the overlaps among the frameworks, BRAVE was used as the reference inventory of broader value elements to build upon and potentially augment.

#### 3.1.1. Identified Value Elements

The ‘Impact Inventory’ (Appendix A) lists the identified value elements. To preserve comprehensiveness, the first four categories of effects from BRAVE were retained. These followed an order from a narrow perspective towards a broader one, and from health-related outcomes towards economic and societal outcomes: narrow (A) and broad (B) health effects were followed by the cost and income outcomes for the healthcare system (C1 and C2), the public sector (C3), and society (D). The narrow category (A) included value elements related to the impact of vaccination on the health of the vaccinated subjects. The broad health effects category (B) included effects related to the impact of vaccination on the health of the unvaccinated subjects and other relevant individuals such as caregivers. Category C was concerned with the impact of vaccination on the health system. Category D focused on the broad societal and economic impact of vaccination beyond the healthcare system, spanning to broad macroeconomic effects. Category E was added based upon the TLR results and especially building on the novel value elements of the ‘ISPOR Value Flower’ [24]. This category of effects included those on the uncertainty of health and economic outcomes, including the psychological impact of vaccination (i.e., value of hope, value of knowing, fear of diseases/contagion), and insurance value.

One of the initial findings from the TLR was that most of the quantifiable value elements and outcomes impacted by COVID-19 were captured in BRAVE. Several elements that were considered relevant to COVID-19 but not explicitly listed in BRAVE were added to the framework, including mental health impact, impact on the health system, effect on public finances, impact on direct costs of non-pharmaceutical interventions (NPIs), impact on foregone education, changes in individual and household behaviour, income equity value, scientific spill-over effects, environmental effects, and uncertainty value. Some of these elements might be implicitly included in BRAVE (e.g., broad health outcomes may include mental health impact), but they were mentioned separately because of their importance and/or prominent consideration in the COVID-19 research. Most of these elements have been primarily associated with the pandemic and the resulting NPIs, as opposed to COVID-19 cases. However, it was also considered that these elements may prove to be of relevance for future immune escape COVID-19 variants, or for other diseases that are associated with large-scale containment measures.

#### 3.1.2. Identified Quantification Methods

The literature review subsequently identified methods to quantify the impact of COVID-19 or vaccination on the value elements that are usually not considered in economic evaluations of vaccines. The amount and strength of the identified evidence and the perceived ability to incorporate the elements into quantitative evaluations varied across value elements. Appendix A summarise these analyses and considerations, whereas Appendix A describes all such identified quantification methods in detail, along with the experts’ views collected during the panel discussions.

Narrow health effects (A) were not in the scope of the TLR, as the impacts on length of life and quality of life (QoL) of patients are usually included in traditional cost-effectiveness analysis of vaccines; as such, several well-known methods exist for the quantification of these effects. The broader health effects (B), including the mental health impact of the COVID-19 pandemic and related social restrictions, the overload on health systems, and the disproportional direct/indirect health burden of disadvantaged groups in society associated with the pandemic, were well documented [39,40,41,42,43,44,45,46]. However, limited evidence was identified to assess the monetary value of vaccination through its impact on mental health or considering the health equity aspects of the pandemic or vaccination [47,48,49]. While certain elements of the public finance impact of the pandemic (C) have been estimated for both the US and UK, no study was identified calculating the full public finance impact or the return on investment of the COVID-19 vaccines [18,50,51].

For the societal and economic effects (D), literature assessing various aspects of the pandemic’s macroeconomic impact was abundant, mainly concentrating on GDP and employment as outcome measures [40,42,52,53,54,55,56,57,58,59,60,61]. Contrarily, limited published evidence was available estimating the impact of vaccination on the direct cost of NPIs [30]. Finally, for uncertainty value (E), increasing COVID-19 vaccination rates have shown psychological benefits, measured by lower levels of anxiety, worry, displeasure, and depression in the US [62]. However, no identified study had attached monetary value to the psychological value of vaccination.

### 3.2. Results of the Expert Elicitations

#### 3.2.1. Panel Characteristics

Panel 1 was composed of a total of nine experts, of whom eight (89%) responded to the individual elicitation in Round 1. The experts had worked in healthcare sector or research for 29 years on an average (min–max: 12–47); most worked in academic or governmental institutions, with seven conducting research primarily in the UK setting, four in the US, and four on an international scale. A total of seven experts participated in Round 2, and a total of nine experts participated in Round 3.

Based on these panel compositions, consensus in Round 1 was defined as >75% (six out of eight) of participant panellists providing either a high rating (4 or 5) or a low rating (1 to 2) to a value element. In Round 2, consensus was defined similarly, using ≥67% (six out of nine) as the threshold.

Given the focus of Panel 2 on the methods to capture broad value, the composition of Panel 2 was narrowed down to the seven experts with health economics expertise that attended Panel 1.

#### 3.2.2. Conceptual Value Framework and Considerations

The experts provided several important conceptual considerations before and after the expert elicitations rounds.

Among the initial remarks, the experts pointed out that the profound impacts of COVID-19 and COVID-19 vaccines showed the limits of existing HTAs for vaccines in capturing their full public benefits. Despite increasing recognition of the broader value of vaccination, narrow direct benefits remain the focus of vaccine HTAs, with potential implications on public investments and immunisation policies. The experts pointed out that the staggering effects of COVID-19 and the broad benefits associated with COVID-19 vaccines provide an opportunity to advance perspectives on the importance of accounting for the full benefits of vaccines in HTAs.

The experts mentioned that, during the identification and collation of elements reflecting the full value of a vaccine, ideally, a framework should integrate elements that are conceptually appropriate for any vaccine evaluation. The relevance and magnitude of impact of each value element of the framework would then differ on a case-by-case basis, depending on the vaccine and settings under assessment. In this regard, the experts agreed that value elements are strongly dependent on the specific vaccine and circumstances it is being evaluated under, specifically considering pandemic versus endemic settings, short versus long analytic time horizons, and direct and indirect effects. For example, the direct costs of NPIs were considered primarily in relation to the pandemic nature of COVID-19 and are less of a concern for endemic phases. Value elements strongly related to NPIs, such as lockdowns, are expected to be highly relevant and have a substantial impact in an evaluation of primary vaccinations in a pandemic setting in the context of low levels of population-based immunity, rising infections, limited vaccine and treatment options, and voluntary and non-voluntary contact restrictions. In an endemic setting, the benefits of vaccination should be balanced with additional factors defining the counterfactual scenario against which the impacts on value elements are expected to be quantified, such as the existence of effective and affordable therapies and the presence of vaccine comparators. However, the pandemic versus endemic differentiation was not considered binary: as COVID-19 transitions to an endemic disease, some of the indirect impacts experienced during the pandemic phase, such as large productivity impacts associated with quarantines, disruptions in production, or foregone education associated with school closures, are still present or have lingering or spill-over effects. The improvements in health brought by the COVID-19 vaccines can reap long-term benefits, including strengthening economic stability, influencing individual and household behaviours such as fertility, and improving educational outcomes. These benefits can be relevant to be considered for future evaluation of boosters and potentially new formulations against new variants, even if with a lower quantitative impact than previously. As such, there was unanimous consensus in recommending that assessments of COVID-19 vaccines in either pandemic or endemic settings should be conducted using a societal perspective.

For the purpose of developing a comprehensive framework of all value elements that, in principle, could co-exist and be considered in vaccine evaluations, the panel agreed to first investigate the conceptual importance of value items that should be part of it. Assessment of the ability to measure these items and incorporate them into an economic evaluation should be a second step. It was emphasised that merely focusing upfront on aspects that are easy to measure and for which good quality evidence is available could lead to neglect of important value elements. In cases where a value element is considered important from a societal perspective, the causal impact of vaccination on it is conceptually established, and the expected value of information is high, stakeholders and the scientific community should be encouraged to refine the methodological toolset and collect data to fill in the evidence gaps. Filling evidence gaps to capture broad value in vaccine assessments requires resources, but the evidence generated may align incentives for the industry and research, and may improve the quality of decision making.

The experts considered it pragmatic to build and expand on BRAVE. While reviewing the list of value elements included in the ‘Impact Inventory’ expanding on BRAVE, the experts pointed out that the elements listed represent mutually dependent concepts. Depending on how they are labelled and described, they can be sometimes interpreted as overlapping. For instance, psychological benefits of vaccination may potentially and partially translate to impact on mental health as measured by incidence of depression and anxiety; productivity impact influences public sector budget through transfers and taxes, and both are related to macroeconomic outcomes; impact on patient QoL is part of the burden of disease. The experts highlighted that, while a comprehensive framework inevitably includes overlapping value elements, assessments of specific vaccines in specific contexts should be designed in a way that excludes overlaps and double counting. It is theoretically possible to limit double counting by presenting evidence on overlapping items separately rather than summing them quantitatively into a single value metric. However, this is arguably not a preferred approach in an HTA context, as it does not lead to an ultimate metric of cost-effectiveness that could be compared against predefined thresholds. The panel recommended that value elements that cannot be quantified and included in the assessment without overlaps should be either considered for exclusion from the analysis or analysed in a qualitative way to support deliberations. However, they emphasised that, in practical applications, overlaps are less likely to arise in assessments based on well-defined integrated epidemiological and economic models, potentially with relatively short analytic time horizons.

The experts pointed out that the assessment of the impact of a disease and the evaluation of a vaccine should balance positive and negative aspects and consider the ‘net’ effect of each value element. In the case of COVID-19, for example, negative externalities of COVID-19’s impact on the health system include diversion of resources, leading to delayed diagnoses and treatments for other diseases, and reduced uptake of other immunisations. On the other hand, some of the positive externalities that could arguably be linked to both COVID-19 vaccination and NPIs include air quality improvement during lockdowns, and impacted transmission patterns and infection rates of other respiratory diseases such as respiratory syncytial virus infection and influenza [63,64,65,66]. Similarly, it was emphasised that the potential negative effects of vaccination should be considered and spelled out separately in the framework. These include disutility and work productivity loss related to adverse events, vaccine anxiety leading to fear and uncertainty over long-term effects of vaccination, and potential negative effects derived from compensatory behavioural adjustments post-vaccination, for example, related to increased social interactions based on the perception of being protected from infection. While attaching a quantitative value to the psychological effects of vaccination was considered difficult, conceptually these aspects were considered to have an impact on uptake, infection numbers and, potentially, quality of life. Moreover, they were considered especially relevant by the panel in mandatory vaccination programs, where some people may have to take the vaccine despite having a negative subjective valuation for it. As a result of these considerations, vaccine anxiety was added under the category E3 of the ‘Impact Inventory’, and the following revisions were made: the mortality and QoL impact of potential adverse events related to vaccination were captured more explicitly in category A1; the attitude towards risk of infection and infection control measures, and willingness to vaccinate against other diseases, were added under the D4 category of behavioural changes; mental health and foregone education were captured more explicitly as well. The value framework is visualised in Figure 3.

#### 3.2.3. Prioritisation of Value Elements for COVID-19 Vaccine Evaluations

The voting results related to the prioritisation of value elements for inclusion in COVID-19 vaccine value assessments based on their conceptual appropriateness are presented in the ‘Priority for Inclusion’ columns in Figure 4. The value elements are listed in the order of their priority for inclusion in assessment, wherein consensually rated elements were highlighted in green, and elements substantially lacking consensus in yellow.

In Rounds 1 and 2 combined, 11 value elements were consensually rated as of ‘high importance’ (rated 4 or 5 on the 5-point Likert scale) for inclusion in COVID-19 vaccine assessments, including three elements from the broader health effects (B), four from the societal and economic impact (D), two from public finances (C), and one each from the insurance value (E) and impact on length of life and quality of life (QoL) of patients (A). The 11 elements were transmission value, burden of disease, avoided care cost of infected patients, macroeconomic effects, impact on length and QoL of patients, impact on patient productivity, impact on foregone education of patients, changes in individual and household behaviour, health system impact, avoided care costs related to broad health effects, and insurance value.

After Round 2, the experts discussed the criteria considered for prioritising value elements, whether the value element is quantifiable in addition to its likely magnitude, whether it is feasible to compare to the cost of vaccination rollouts, and the consideration of a combination of what matters to people, governments, and what is known about the impact of COVID-19.

The voting results related to subsequent questions on the quality of evidence and the feasibility of inclusion in economic evaluations are presented in the corresponding columns in Figure 4 and Appendix A. Most of the value elements prioritised for inclusion in HTA evaluations by the experts were characterised by high quality and high feasibility. Figure 4 shows that, among all items prioritised for inclusion, the ones that were consensually rated as ‘high’ and also backed up by ‘high-quality evidence’ and a ‘high feasibility for inclusion’ were the ones that were already included in many or all vaccine value assessments: transmission value, burden of disease, avoided care cost of infected patients, and impact on length and QoL of patients. Insurance value was prioritised in Round 2, although with a relatively low score for evidence and feasibility. High priority and feasibility items without consensus on high evidence included impact on patient productivity and changes in individual and household behaviour. The experts discussed many aspects related to individual and household behaviour, such as decisions related to fertility, consumption, leisure, savings, living arrangements, attitudes towards risk of infection, willingness to vaccinate against other diseases, and impact on quality of life related to isolation at home. Some of these aspects are captured via other value elements in the framework (e.g., impact on savings under macroeconomic effect). In those instances, behavioural changes constitute a channel through which value elements are impacted by COVID-19 and vaccination.

One aspect that was considered especially important was the prevalence of elastic demand for prevention, whereby increased vaccination uptake in the population could lead to lower individual risk-avoiding behaviour. Increased risk-taking following vaccination may lead to more social contact and potentially more infections than what would have been observed without the behavioural response.

Value elements that had consistently ‘low’ (rated 1 or 2) scores for priority of inclusion were environmental effects, value of knowing, value of hope, and scientific spill-over effects. For the first three, this assessment was coupled with a consensually low rating for quality of evidence, while environmental effects also received a low rating for feasibility of inclusion.

Besides the 11 value elements prioritised in Rounds 1 and 2, during Round 3, six out of nine participants rated either ‘impact on QoL of carers’ or ‘impact on productivity of other individuals’ as being one of the top five elements among the ones lacking consensus in Rounds 1 and 2, bringing the total number of prioritised value elements to 13. Two value elements were not consensually prioritised but were emphasised as not receiving sufficient attention in current assessments: impact on caregiver productivity, and health equity and related aspects of social and economic equity. From the patient’s perspective, it was highlighted that informal carers may face both productivity and health impacts as a result of their caring responsibilities. A vaccine that reduces the need for informal care can generate economic savings related to unpaid care, and may contribute to overall health equity.

During the discussions, it was highlighted that while antibiotic usage has been impacted by COVID-19, the significance and impact of this effect on the development and transmission of AMR was not yet measured. The limited evidence contributed to a relatively low overall rating for ‘AMR prevention’.

#### 3.2.4. Evaluation of Identified Quantification Methods

The review of quantification methods focused on selected value elements prioritised for inclusion in COVID-19 vaccine evaluations. The Panel 2 discussion prioritised the review of methods for the following value elements: impact on work productivity, macroeconomic effects, impact on foregone education and health system impacts. Appendix A presents the results from Round 1 voting on the appropriateness of the identified quantification methods for the elements that were considered relevant by Panel 1.

The list of methods that were identified included well-established approaches for quantifying impact on work productivity, namely the friction cost and the human capital methods [67]. The experts discussed the pros and cons of both and commented that the human capital approach captures lost income due to mortality/morbidity associated with a disease at an individual level, and it is practical for diseases leading to short-term sickness absences such as acute respiratory diseases. However, some of the aspects of productivity are not generally considered in these types of assessments, such as the fact that losing one’s job and becoming re-employed is associated with a substantial loss of firm-specific human capital. Aligned with prior observations related to the need to balance positive and negative effects, the experts pointed out that the productivity loss linked to the administration and management of adverse events of vaccination should be accounted for, too. Finally, the experts also highlighted that, in a pandemic setting, estimation of the productivity impact needs to consider individuals beyond patients and caregivers. The effect of mandatory quarantine periods and the higher risk of longer leave and unemployment may be mitigated by recent adjustment to new working arrangements and cost savings arising from avoiding commuting to the office.

The macroeconomic impact emerged as a key topic in the panel discussions and recent literature [13,26]. The TLR identified a long list of mutually dependent macroeconomic aggregates and indicators. Some of the quantified outcomes included tourism income, production, GDP in total and at the sectoral level, GDP components (e.g., consumption, household income), employment including hours worked (partially overlapping with ‘productivity effect’ as included in HTA), domestic and foreign investment, financial market indicators (e.g., stock market indices), and societal welfare measures. The experts pointed to existing research focusing on GDP estimates [40], and highlighted that, when assessing the macroeconomic impact of vaccines through avoiding NPIs, focusing on the aggregate GDP measure (or gross value added) as the only macroeconomic outcome measure was considered an appropriate approach for synthesising macroeconomic impact. This single measure avoids the need to look at multiple interrelated macroeconomic outcomes, minimising the risk of double counting. However, as GDP is not sensitive to distributional outcomes, impact on health equity and income inequalities through social welfare measures were also recommended. Building de novo macroeconomic or combined epidemiological and macroeconomic models can be helpful from a scenario analysis perspective to assess whether results point in the same direction, but such complex models do not seem appropriate for HTA purposes and for endemic settings, due to their complexity, data needs, and inherent uncertainties.

The economic impact of education loss has been assessed in terms of lost schooldays (due to illness and/or school closures), which have been linked to test scores, lost future income and GDP loss. The Organisation for Economic Co-operation and Development (OECD) approach for assessing the impact of lost education on individual income and GDP was considered straightforward and worthy of conduction [56]. However, the experts highlighted that the method is not sensitive to distributional consequences, which were considered important. The impact of lost schooling on the distribution of test scores is relevant both from an equity perspective and because the share of high achievers is a key driver in economic growth. The panel suggested an assessment of impact across different education levels. From a technical perspective, the experts mentioned that such assessments should consider the effects of policy measures aiming to mitigate the impact of school closures on cognitive outcomes, such as alternative teaching methods (summer camps, online programs). Microsimulation models estimating the impact of lost education on individuals’ future productivity [58] were considered complex from a HTA perspective.

Multiple sources documented the impact of the COVID-19 pandemic stretching the resilience of health systems to provide non-COVID-19-related care in the same manner as the pre-pandemic level. These effects have been measured by simple pre- and post-March 2020 comparisons in time series with a variety of outcome variables such as length of waiting lists, length of waiting times, number of treatments and screenings performed (e.g., for cancer), hidden needs (an estimate on the number of people that need care but have not yet come forward to receive care), number of incomplete patient pathways, and excess deaths [43,46]. The experts considered it appropriate to evaluate healthcare resources by their opportunity cost, as performed by Brassel et al. [18], as opposed to their accounting cost (e.g., cost of a hospital bed). Brassel et al. [18] estimated the opportunity cost of treating one patient instead of another in an excess demand situation. For assessing the monetary value to the health system, the opportunity costs were proxied by the net monetary benefit foregone by treating a vaccine-preventable outcome instead of treating a patient from the waiting list. The experts discussed alternative methods for estimating the value of non-COVID-19 life years saved using excess deaths from non-COVID-19 causes and the value of a life year [42], or, alternatively, of a statistical life. They discussed that excess deaths as a measure for the indirect mortality impact of COVID-19—including but not limited to the health system impact—can be controversial, and should only be assumed during time periods when demand for health services exceeds capacity. Hence, in forward-looking analyses, for instance, a multiplier capturing the relationship between the number of intensive care unit (ICU) cases and excess inpatient deaths could be used for predicting excess deaths based on case numbers. The experts also pointed out that, ideally, the applications of this method should consider interdependencies across care settings and the underlying background risks of existing conditions in the population of interest. For example, missed check-ups or delayed treatments for conditions treated in the primary care sector may result in deterioration and accelerated increase in severity of such conditions that may ultimately require inpatient care. Finally, the experts suggested exploring assessments of QoL decrements as well, especially for deteriorating conditions that are unlikely to cause short-term death but can cause suffering to unattended patients waiting to receive care.

The experts highlighted challenges in measuring the impact of individual and household behaviour. They pointed out that this category may include a long list of poorly documented and quantifiable effects related to ‘household production’, for example, increased time devoted to household childcare responsibilities as a result of school closures.

A broad economic evaluation can capture health, economic, and social impacts. As a direction for future research in this field, the experts commented that, when structuring the economic assessment of a vaccine, an integrated epidemiological, economic and social model, despite being complex for standard HTA, can help with conceptualising the framework and evaluating the relationship among the value elements. They highlighted that, when presenting evidence on the value of vaccination, it is important to assess both parameter uncertainty and model uncertainty by pursuing alternative modelling approaches and presenting their comparative results transparently. Whenever possible, comparing conclusions from trial-data-based estimates against real world evidence is recommended. When considering the inclusion of conceptually important but non-monetary value items in economic evaluations, they suggested conducting a multi-criteria decision analysis with expert involvement. Finally, conducting comparisons of vaccine value estimates based on evidence-based narrow versus broader perspectives would provide insights into the value of information to be potentially generated for future assessments.

## 4. Discussion

The COVID-19 pandemic and its profound impact on society have reignited the debate about the scope of vaccine value assessments. This research work aimed to contribute to this discussion and advance the growing consensus that vaccines generate value beyond direct effects for vaccinated individuals and healthcare systems only. Using COVID-19 as a case study and leveraging a mixed-method approach combining a TLR and expert elicitation, this research work introduced a comprehensive list of value elements associated with vaccination for potential consideration either in endemic or pandemic settings. When applying the framework to COVID-19 vaccines in a post-pandemic setting, 13 value elements were consensually rated highly important by the experts for consideration in HTA evaluations.

The expert consultation process resulted in a clear consensus on the need to expand existing narrow vaccine HTA frameworks to better emphasise the value of vaccination from a broad societal perspective. The experts discussed the need for a framework that captures value elements reflecting the impact of vaccination on patients, caregivers, and rest of the society. For practical applications, this framework should include value elements that are potentially operationalisable and likely quantifiable, but should not necessarily be confined to items for which good-quality quantitative evidence is already available. The conceptual and quantitative importance of value elements is expected to vary across diseases, country settings (private, public), endemic and pandemic context, and the availability of treatments and alternative vaccine platforms.

The resulting framework (Figure 4) was based on both vaccine- and non-vaccine-specific frameworks and showed similarities with previous studies, as several of the value elements prioritised for broader recognition were discussed in expert panels by Bell et al. [13], Postma et al. [26], and Asukai et al. [14] with the latter focusing on COVID-19 therapeutics. High-rated value elements not currently considered in vaccine evaluations but covered in the BRAVE framework included macroeconomic effects and cost consequences of health system impact. Value elements not included in the BRAVE framework but rated highly by experts were impact on foregone education of patients, health (QoL) consequences of health system impact, insurance value, and changes in individual and household behaviour—specifically, the tendency of people to engage in more social contact as a response to decreased infection risk due to vaccination. Some of these value elements were recommended for prioritisation in Postma et al. [26], such as macroeconomic impact, health system externalities, and foregone education, which was covered under productivity impact. Asukai et al. [14] discussed equity, disease severity, insurance value, scientific and family spill-over. Environmental impact—or, in Postma et al. more specifically, carbon footprint—received low scores in both expert groups. Psychological effects of vaccination and scientific spill-overs were valued less in this study’s expert panels than in the analysis reported by Postma et al. [26]. Similarly, while some experts emphasised the importance of considering the equity aspect of vaccination, there was no consensus on its prioritisation in this expert panel, contrary to its high support in Postma et al. [26] and Asukai et al. [14] in a pandemic setting. Furthermore, although AMR prevention (B5) was rated generally low based on availability and quality of evidence, some of the experts emphasised its quantification feasibility and potential future value, should more evidence emerge.

The framework proposes new value elements that have not been previously covered, such as vaccine anxiety, health, QoL, and economic impact on individuals other than patients and caregivers. The relevance, quality of evidence and possibility of inclusion of the value elements in future vaccine evaluations were assessed by the experts, with a focus on COVID-19 vaccines in a post-pandemic setting. Conceptual considerations regarding including new value elements in assessments were also discussed, including the importance of generating theoretical and empirical evidence related to value items that are highly relevant but are currently overlooked due to evidence gaps and the lack of appropriate quantification methods.

Differently from prior studies [13,14], this study collected and reviewed currently available quantification methods to assign value to the identified elements. The strengths and limitations of quantification methods were discussed with the experts, both from a general, academic perspective and in the context of HTA evaluations. The conceptual and methodological challenges related to the strong relationship between value elements and resulting overlaps were analysed, and recommendations were made regarding potential approaches to limit double counting.

The results of this study should be considered in the context of several limitations.

As emerged during the panel discussions, several aspects of the framework should be interpreted with caution. The framework brings together value elements identified in the TLR and in different existing vaccine and non-vaccine frameworks where each value element has a different level of detail, validation and tangibility. Seen altogether, the value elements are not mutually exclusive, as several of them are interdependent and overlapping; hence, they should not be seen simply as additive to one another. Further research should focus on further improving their descriptions and delineating their interdependencies. Quite similarly, many of the methods identified by the TLR were considered relevant and suitable for capturing the impacts of vaccines by the panel, although several challenges exist for their use and there is scope for improving them. In addition, the conceptual and quantitative importance of value elements may vary across vaccines, diseases, settings and time horizons. This framework should be used and adapted on a case-by-case basis to identify value elements that are applicable to the specific country and intervention under evaluation. Furthermore, this study focuses on the need for a broad perspective on Vaccine HTAs; however, similar criteria would apply to other health interventions (e.g., pharmaceutical drugs, medical devices) and non-health interventions with health implications, while prioritising the allocation of health budgets.

The composition of our panel mainly represented the US and UK and, despite its diversity, it cannot provide full representation of societal preferences. Hence, the results of this research may not be generalisable to countries with different or less established HTA processes and should be interpreted in the context of the limitations of the panel composition and size. While the response rates across the three rounds were relatively high (88% in Round 1, 77% in Round 2, 100% in Round 3), the stability of responses might have been affected by the differences among participants and their own interpretations of the topics, especially in the early rounds.

Fundamentally, the size of a Delphi panel can range from 3 to 80 participants [34,35,36], and our panel selection criteria and size were consistent with similar prior efforts (Bell et al. [13] included 10 experts in the panel). While we feel that the group of experts was heterogenous and highly knowledgeable in this research field, we acknowledge that any small-sample qualitative research study has limited generalisability. It is possible that a larger panel and/or a different composition could have led to a different final set of recommendations. For example, the results would probably have been different if the panel included a larger number of HTA specialists and a different mix of experiences in the topic of this research work. Furthermore, while the lack of full anonymity between panellists contradicts one of the basic rules in the Delphi method, a lack of discussion could also hamper clarification of disagreements. Hence, our Delphi was modified to include communication among experts and, to minimise the biases from dominance or group pressure, survey responses were always kept anonymous.

Finally, given the large and growing amount of literature, we took a pragmatic approach and focused efforts on a targeted literature review. A substantial amount of evidence was available on the impact of COVID-19 on indirect health outcomes, loss of schooldays, macroeconomic impact, public finances, antibiotic use, and on certain environmental outcomes; however, research evaluating certain value elements and evidence directly relating COVID-19 vaccination to these broad outcomes was scarce and relatively uncertain. As evidence on the impacts of COVID-19 and the COVID-19 vaccines is still accumulating, and methods for quantifying the effects are constantly evolving, further research and expert debate is warranted for this important and complex topic.

COVID-19 imposed devastating health, economic, and social burdens, with lingering consequences on societies and economies. The positive benefits of the global vaccination efforts have been well documented and have highlighted the direct and indirect benefits of COVID-19 vaccines. However, COVID-19 vaccination has not yet reached all who could benefit. To the extent that vaccine hesitancy may persist, it may reduce the aggregate benefits for populations and communities. Moreover, there is no guarantee that the SARS-CoV-2 virus will disappear, as new variants and sub-lineages may continue to emerge. The uncertain future dynamics support continuous pandemic preparedness efforts to ensure rapid and broad access to safe and effective vaccines, therapeutics and diagnostics. 

## 5. Conclusions

The impact of vaccination extends beyond the direct health effects on vaccinated individuals and healthcare systems. Using COVID-19 as a case study and a mixed-method approach, this research work introduces a conceptual framework of elements to consider when assessing the value of vaccines.

From the exercise emerged an unequivocal consensus on the importance of assessing the value of COVID-19 vaccines using a societal perspective. Several value elements were consensually rated highly important by the experts for consideration in HTA evaluations of COVID-19 vaccines in a post-pandemic setting. Moreover, recommendations and challenges on methods to quantify these elements are summarised. The findings of this research and the lessons from COVID-19 create opportunities to advance considerations on the incorporation of the full effects of vaccines and other health-protecting and health-promoting interventions in HTAs.

## Figures and Tables

**Figure 1 vaccines-11-00234-f001:**
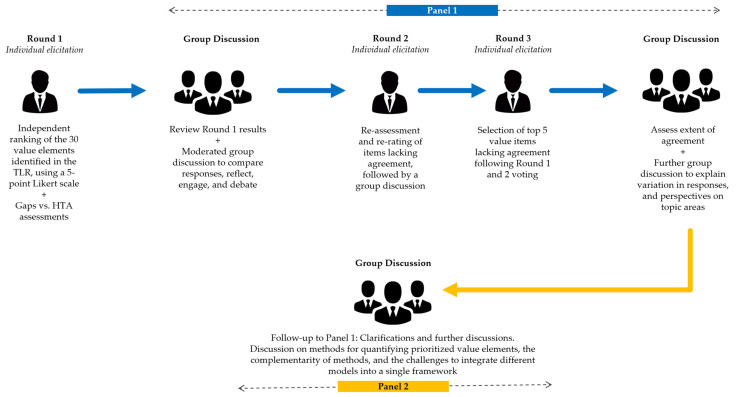
Expert elicitation using a modified Delphi method. HTA, health technology assessment; TLR, targeted literature review.

**Figure 2 vaccines-11-00234-f002:**
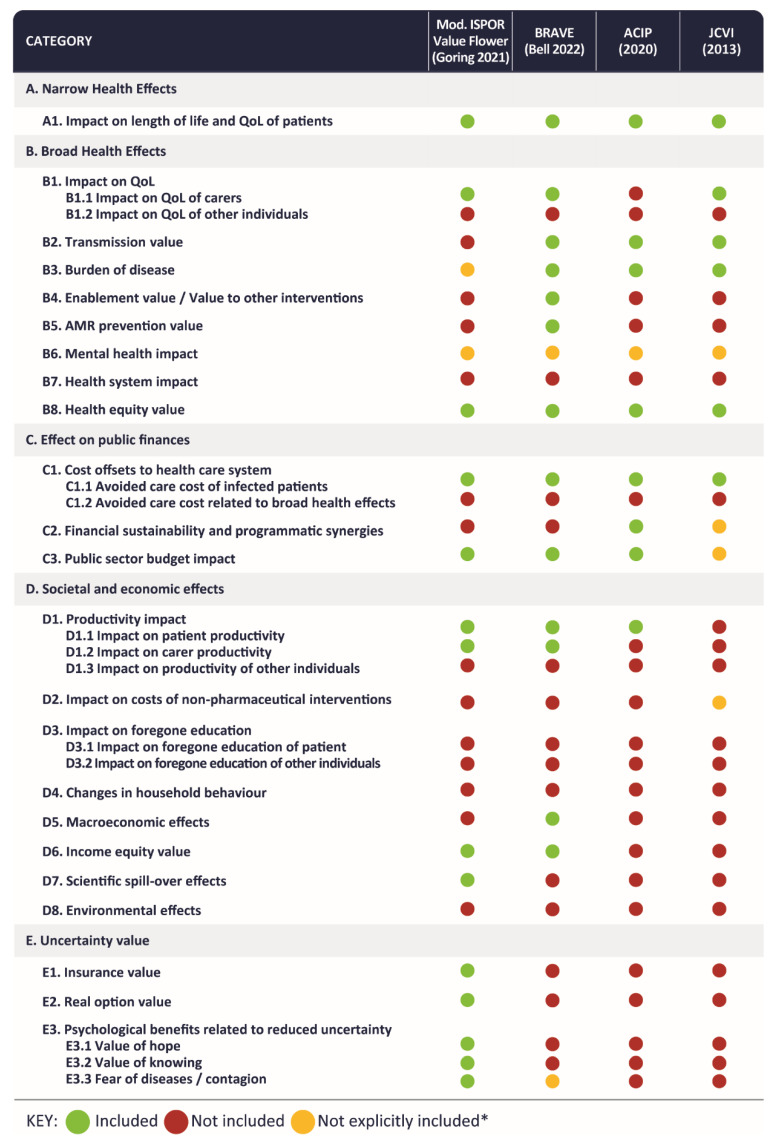
Inclusion of value elements in key value frameworks. ACIP, Advisory Committee on Immunization Practices (US); BRAVE, Broader Value of Vaccines; JCVI, Joint Committee on Vaccination and Immunisation (UK); QOL, Quality of Life; * Not explicitly mentioned but may be considered as part of included value elements in the framework, or formally or informally considered in some assessments by HTA body (ACIP, JCVI); Note: Vaccine Anxiety was added to the framework ex post based on the panel discussions.

**Figure 3 vaccines-11-00234-f003:**
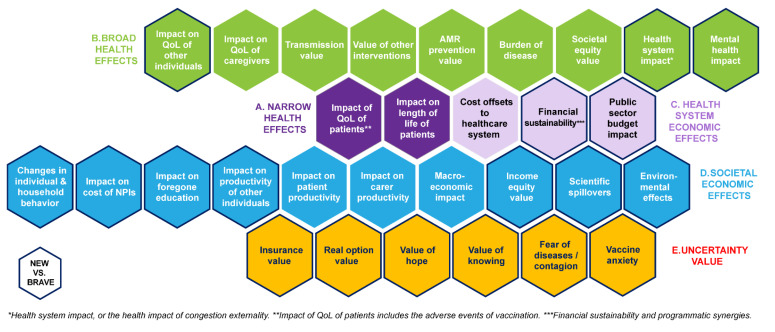
Visualised Vaccine Value Framework.

**Figure 4 vaccines-11-00234-f004:**
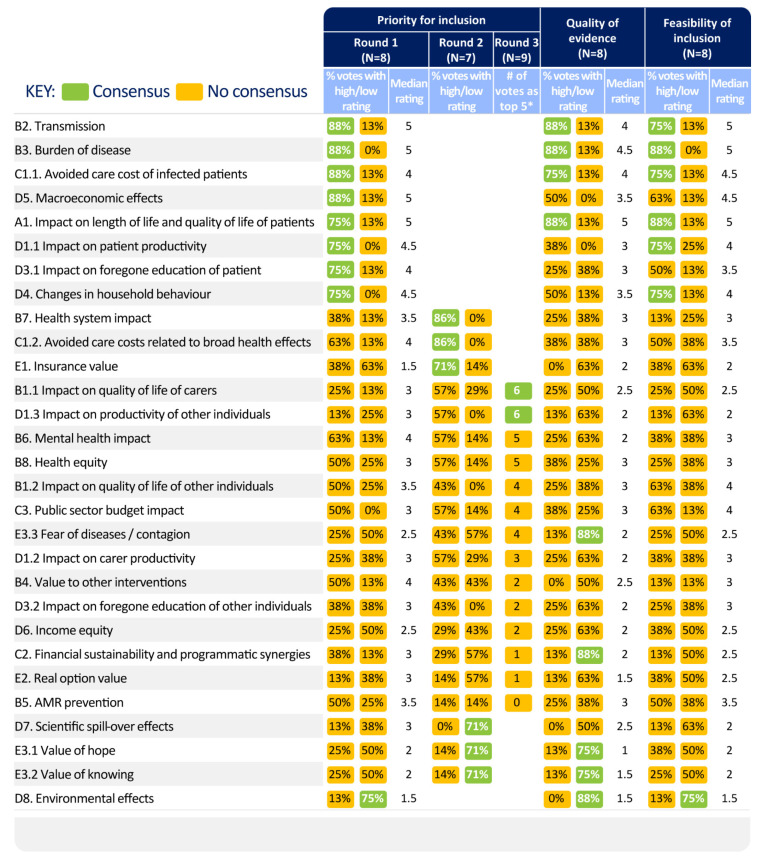
Results of expert elicitation rounds. * Experts were asked to choose the top five elements among the ones lacking consensus in Rounds 1 and 2. High rating: 4 or 5 on a 5-point Likert scale. Low rating: 1 or 2 on a 5-point Likert scale. Consensus in Round 1 was defined as >75% (six out of eight) of participants providing either a high rating (4 or 5) or a low rating (1 to 2) for a value element. In Round 2, consensus was defined similarly, using ≥ 67% (six out of nine) as threshold. Note—Vaccine anxiety was added to the framework as a result of expert panel discussions and is not covered in this table.

## Data Availability

Aggregate data generated or analysed during this study are available from the corresponding author upon review.

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
