# Peer review of "The Societal Value of Vaccines: Expert-Based Conceptual Framework and Methods Using COVID-19 Vaccines as a Case Study"

_vaccines, 2023, doi:10.3390/vaccines11020234_

Round 1

Reviewer 1 Report

The paper is devoted to the development of methodology of the assessment of vaccination-associated issues for potential consideration either in endemic or pandemic settings. The objectives of the works are declared as (1) to create a consensus-based conceptual framework capturing the broader value of vaccination, (2) to gather expert recommendations and views on challenges related to analytical approaches measuring these benefits, (3) to assess the applicability of the framework and the identified methods for evaluations of COVID-19 vaccines, with a focus on post-pandemic settings. Suggested methodology consists of two main stages: a targeted literature review and expert panels.

In the application to COVID-19 vaccines in a post-pandemic setting, 11 elements were suggested as most important: transmission value, burden of disease, avoided care cost of infected patients, macroeconomic effects, impact on length and QoL of patients, impact on patient productivity, impact on foregone education of patients, changes in individual and household behavior, health system impact, avoided care costs related to broad health effects, and insurance value.

COVID-19 pandemic showed, especially in the beginning, that the society is amazingly unprepared for such large-scale events.  This work and the related literature are aimed to overcome this unpreparedness. Some questions, which seem to be important in the context of vaccination during epidemics of respiratory viral infections, require more attention. Among them:

11.      Anti-vaccine attitude was quite strong during COVID-19 vaccination campaign. It is important to clarify how and why it emerges, how to deal with it, the role of mass media and social networks, and so on.

22.       The second objective of the work concerns expert recommendations. It is not clear how this objective is reached and whether this question is really in the scope of this work. In particular, what are the recommendations in the case of COVID-19:  how to limit macroeconomic effects; how healthcare should be organized in the case of large-scale epidemics; how to provide more efficient and safe vaccine development, and so on.

33.       How vaccination-associated assessment should be adapted to different countries and regions depending on their economic development and societal aspects such as cultural traditions, religion, and so on.

A minor remark, 11 elements are indicated as most import for COVID-19 on line 398 and 13 elements on line 573. Check please.

Reviewer 2 Report

The authors present a very interesting manuscript, well written and with relevant information. The authors make an adequate introduction to the state of the art, conceptually correct and that manages to justify the main aim of the study. In this sense, I would suggest that the authors place more emphasis on the aspects of possible geospatial differences.

The methodology is clear and adequately described. I suggest to the authors, improve the quality of the figures in this section, making the explanation of the figure legends more self-explanatory.

The results are adequately written and with emphasis on what is important. However, the figures are of low quality and need to be improved and further explained and made more self-explanatory.

The discussion is accurate, but limitations, translation, and future dynamics must be justified and discussed.

The conclusions are adequately established on the results.

A specific graphic summary would be necessary.
